# Essential Oils from Vietnamese Asteraceae for Environmentally Friendly Control of *Aedes* Mosquitoes

**DOI:** 10.3390/molecules27227961

**Published:** 2022-11-17

**Authors:** Tran Minh Hoi, Prabodh Satyal, Le Thi Huong, Dang Viet Hau, Tran Duc Binh, Dang Thi Hong Duyen, Do Ngoc Dai, Ngo Gia Huy, Hoang Van Chinh, Vo Van Hoa, Nguyen Huy Hung, William N. Setzer

**Affiliations:** 1Department of Plant Resources, Institute of Ecology and Biological Resources, Vietnam Academy of Science and Technology, Hanoi 100000, Vietnam; 2Aromatic Plant Research Center, Lehi, UT 84043, USA; 3School of Natural Science Education, Vinh University, Vinh City 43000, Vietnam; 4Center for Research and Technology Transfer, Vietnam Academy of Science and Technology, Hanoi 100000, Vietnam; 5Faculty of Agriculture, Forestry and Fishery, Nghe An College of Economics, Vinh City 43000, Vietnam; 6Center for Advanced Chemistry, Institute of Research and Development, Duy Tan University, Da Nang 550000, Vietnam; 7Faculty of Natural Sciences, Hong Duc University, Thanh Hoa 440000, Vietnam; 8Department of Pharmacy, Duy Tan University, Da Nang 550000, Vietnam; 9Department of Chemistry, University of Alabama in Huntsville, Huntsville, AL 35899, USA

**Keywords:** Blumea lacera, Blumea sinuata, Emilia sonchifolia, Parthenium hysterophorus, Sphaeranthus africanus, essential oil composition, larvicidal activity

## Abstract

Mosquitoes, in addition to being a biting nuisance, are vectors of several pathogenic viruses and parasites. As a continuation of our work identifying abundant and/or invasive plant species in Vietnam for use as ecologically friendly pesticidal agents, we obtained the essential oils of *Blumea lacera*, *Blumea sinuata*, *Emilia sonchifolia*, *Parthenium hysterophorus*, and *Sphaeranthus africanus*; analyzed the essential oils using gas chromatographic techniques; and screened the essential oils for mosquito larvicidal activity against *Aedes aegypti* and *Aedes albopictus*. The most active larvicidal essential oils were *B. sinuata*, which was rich in thymohydroquinone dimethyl ether (29.4%), (*E*)-β-caryophyllene (19.7%), α-pinene (8.8%), germacrene D (7.8%), and α-humulene (4.3%), (24-h LC_50_ 23.4 and 29.1 μg/mL) on *Ae. aegypti* and *Ae. albopictus*, respectively, and *Emilia sonchifolia*, dominated by 1-undecene (41.9%) and germacrene D (11.0%), (24-h LC_50_ 30.1 and 29.6 μg/mL) on the two mosquito species. The essential oils of *P. hysterophorus* and *S. africanus* were also active against mosquito larvae. Notably, *B. sinuata*, *P. hysterophorus*, and *S. africanus* essential oils were not toxic to the non-target water bug, *Diplonychus rusticus*. However, *E. sonchifolia* essential oil showed insecticidal activity (24-h LC_50_ 48.1 μg/mL) on *D. rusticus*. Based on these results, *B. sinuata*, *P. hysterophorus*, and *S. africanus* essential oils appear promising for further investigations.

## 1. Introduction

The Asteraceae is the largest family of flora in the world, comprising about 1550 genera and about 23,000 species [1]. In Vietnam, there are about 126 genera and 379 species from this family [2]. Many species are used as medicines, for isolation of essential oils, or as ornamentals [2].

*Blumea lacera* (Burm. f.) DC. (syn. *Conyza lacera* Burm. f., *Blumea bodinieri* Vaniot, *Blumea dregeanoides* Sch. Bip. ex A. Rich., *Blumea duclouxii* Vaniot, *Blumea glandulosa* DC., *Blumea subcapitata* DC., *Blumea velutina* (H. Lév. and Vaniot) H. Lév. and Vaniot, *Conyza velutina* H. Lév., and *Senecio velutinus* H. Lév. and Vaniot) is found in China, Bhutan, India, Japan, Laos, Malaysia, Myanmar, Nepal, New Guinea, Pakistan, Sri Lanka, Thailand, and Vietnam [1]. The pharmacognosy and phytochemistry of *B. lacera* have been reviewed [3]. In traditional medicine, *B. lacera* has been used as an expectorant, diuretic, astringent, antispasmodic, antipyretic, antioxidant, antidiarrheal, liver tonic, and stimulant [4]. The leaves of *B. lacera* are fragrant, and in Vietnam, are used as a vegetable as well as a medicine to treat boils and stop bleeding [5].

*Blumea sinuata* (Lour.) Merr. (syn. *Blumea laciniata* (Wall. ex Roxb.) DC., *Conyza laciniata* Roxb., Asteraceae) [6] is native to southern China, India, Pakistan, Sri Lanka, Bhutan, Nepal, Myanmar, Malaysia, Indonesia, the Philippines, and Vietnam, naturally ranging from southern China and India, south through Indonesia, Malaysia, Myanmar, Thailand, and Vietnam [1]. Its leaves and stems are used to treat boils, remove toxins from the body, and stop bleeding. Leaves of *B. sinuata* have been used to treat influenza, rheumatism, bone pain, or pain due to injury or swelling [5]. A review of the medicinal chemistry, phytochemistry, and pharmacology of the *Blumea* genus has been published [7].

*Emilia sonchifolia* (L.) DC. (syn. *Cacalia sonchifolia* L., *Crassocephalum sonchifolium* Less., *Emilia mucronata* Wall., *Emilia purpurea* Cass., *Emilia rigidula* DC*., Emilia scabra* DC., *Emilia sinica* Miq., *Senecio ecalyculatus* Sch. Bip., *Senecio rapae* F. Br., *Senecio sonchifolius* Moench) is a pantropical weed of Old World origin [8]. Ethnomedically, the plant has been used to treat eye sores, convulsion, cuts, wounds, rheumatism, and insect bites [9]. In Vietnam, the leaves and young tops are used as vegetables, and the whole plant is used as medicine to reduce fever [5].

*Parthenium hysterophorus* L. (syn. *Argyrochaeta bipinnatifida* Cav., *Argyrochaeta parviflora* Cav., *Echetrosis pentasperma* Phil., *Parthenium lobatum* Buckley, *Parthenium pinnatifidum* Stokes) is believed to be native to the Gulf of Mexico, including Honduras, Guatemala, and Mexico, as well as the West Indies [10]. The plant was introduced to Australia, India, southern China, and Vietnam, where it became a noxious weed [11,12]. Nevertheless, the plant has shown potential medicinal applications [13,14,15].

*Sphaeranthus africanus* L. (syn. *Sphaeranthus cochinchinensis* Lour., *Sphaeranthus glaber* DC., *Sphaeranthus globosus* Wall. ex DC., *Sphaeranthus hildebrandtii* Baker, *Sphaeranthus indicus* Kurz, *Sphaeranthus laevigatus* Wall. ex DC., *Sphaeranthus microcephalus* Vatke, *Sphaeranthus microcephalus* Willd., *Sphaeranthus ovalis* Steetz, *Sphaeranthus paniculatus* Cass., *Sphaeranthus sphenocleoides* Oliv. and Hiern, *Sphaeranthus suberiflorus* Hayata) is native to Africa (Kenya, Tanzania, Mozambique, and Madagascar), tropical Asia (Bangladesh, Borneo, Cambodia, south-central and southeastern China, Hainan, India, Malaya, Myanmar, Nepal, Philippines, Sri Lanka, Taiwan, Thailand, and Vietnam), and Australia (Northern Territory, Queensland, and Western Australia) [16]. In Vietnam, a decoction of the leaves of *S. africanus* is used to prepare a mouthwash to treat sore throats [5]. Several biologically active carvotacetones have been isolated from *S. africanus* extracts [17,18,19].

*Aedes* mosquitoes (Culicidae) are acknowledged vectors of numerous pathogenic viruses. *Aedes aegypti* (L.) is known to transmit the yellow fever, Zika, dengue, and chikungunya viruses [20], whereas *Aedes albopictus* (Skuse) is a vector for West Nile, Japanese encephalitis, and Eastern equine encephalitis, as well as dengue and chikungunya viral pathogens [21]. Dengue fever is widespread in Southeast Asia, including Vietnam, and causes considerable health and economic burden [22]. Both chikungunya [23] and Zika [24] viral infections are emerging diseases in the region. Though synthetic insecticides have been used to control mosquito populations, there is growing concern regarding insecticidal resistance [25,26], environmental degradation [27,28], and harm to non-target organisms [29,30]. Essential oils have been recognized as potential alternatives to synthetic insecticides for control of insect pests, including mosquitoes [31,32].

As part of our research into the identification of readily available native and invasive plants in Vietnam as sources of essential oils for ecologically friendly pest control agents [33,34,35,36], we investigated *B. lacera*, *B. sinuata*, *E. sonchifolia*, *P. hysterophorus*, and *S. africanus* essential oils for mosquito larvicidal activity against *Aedes aegypti* (L.) and *Aedes albopictus* (Skuse) (Diptera: Culicidae) mosquitoes. These species of mosquitoes are the principal vectors of the dengue fever virus in Vietnam [37]. To test selectivity, we screened the essential oils against the non-target water bug, *Diplonychus rusticus* (Fabricius), a predator of mosquito larvae. There have been several reviews on the potential pesticidal utility of essential oils to control mosquito populations [38,39,40,41].

## 2. Results and Discussion

### 2.1. Essential Oil Compositions

#### 2.1.1. *Blumea lacera*

Floral, leaf, and stem essential oils of *B. lacera* were obtained at 1.10, 1.56, and 0.35%, respectively. The chemical compositions of *B. lacera* essential oils are presented in Table 1. The most abundant chemical components in the essential oils of *B. lacera* were (*E*)-β-caryophyllene (23.8, 27.2, and 11.7%), germacrene D (18.5, 21.0, and 11.2%), thymohydroquinone dimethyl ether (5.0, 4.1, and 28.4%), γ-curcumene (5.9, 7.7, and 4.7%), *ar*-curcumene (8.0, 3.7, and 1.9%), and α-zingiberene (4.7, 7.1, and 4.6%) in the flowers, leaves, and stems, respectively.

A *B. lacera* leaf essential oil sample from Idaban, Nigeria, was found to contain thymohydroquinone dimethyl ether (33.9%) and (*E*)-β-caryophyllene (10.7%) as major components [42]. Similarly, the two essential oil samples from aerial parts of *B. lacera* from central Vietnam were rich in (*E*)-β-caryophyllene (12.0 and 8.3%), thymohydroquinone dimethyl ether (11.4 and 6.6%), and caryophyllene oxide (21.7 and 11.9%) [43]. Joshi and co-workers have noted large variations in essential oil compositions in samples from different geographical regions of India with thymohydroquinone dimethyl ether ranging from 0.4 to 28.7% and (*E*)-β-caryophyllene from 0.5 to 25.5% [44]. In contrast, a previous examination of the essential oil from the aerial parts of *B. lacera* from Biratnagar, Nepal, found the oil to be dominated by (*Z*)-lachnophyllum ester (25.5%), (*Z*)-lachnophyllic acid (17.0%), germacrene D (11.0%), (*E*)-β-farnesene (10.1%), bicyclogermacrene (5.2%), (*E*)-caryophyllene (4.8%), and (*E*)-nerolidol (4.2%) [45]. Both the essential oil and (*Z*)-lachnophyllum ester showed cytotoxic, antibacterial, and antifungal activity. Interestingly, neither lachnophyllum esters nor lachnophyllic acids were detected in the essential oils from Vietnam. It is not clear what factors contribute to the large variations in essential oil compositions, but environmental influences (climate, altitude, latitude, and edaphic conditions), seasonality, phenology, genotype variation, or extraction method have often been attributed to rationalize essential oil compositional differences [46].

#### 2.1.2. *Blumea sinuata*

The fresh aerial parts of *B. sinuata* were hydrodistilled using a Clevenger apparatus to obtain the essential oil in 0.16% yield. The essential oil composition of *B. sinuata* is shown in Table 2. The major components in the essential oil of *B. sinuata* were thymohydroquinone dimethyl ether (29.4%), (*E*)-β-caryophyllene (19.7%), α-pinene (8.8%), germacrene D (7.8%), and α-humulene (4.3%). As far as we are aware, there is only one previous report on the essential oil of *B. sinuata* (as *B. laciniata*, from Dapoli region, Maharashtra, India) [47]. The GC–MS analysis, however, is not reliable, so a meaningful comparison of the compositions is not possible.

#### 2.1.3. *Emilia sonchifolia*

Hydrodistillation of the fresh aerial parts of *E. sonchifolia* gave a 0.51% yield of essential oil. A total of 43 compounds were identified, accounting for 93.2% of the total composition (see Table 3). Gas chromatographic analysis of *E. sonchifolia* essential oils revealed the oil to be dominated by 1-undecene (41.9%) and germacrene D (11.0%). The essential oil composition of *E. sonchifolia* from Vietnam is in marked contrast to the essential oils from Belagavi, Karnataka, India [48] or Ojo State, Nigeria [49]. The *E. sonchifolia* sample from India was rich in the sesquiterpene hydrocarbons, (*E*)-β-caryophyllene (22.7%) and γ-muurolene (32.1%). The essential oil from Nigeria was also rich in sesquiterpene hydrocarbons, namely (*E*)-β-caryophyllene (15.7%), γ-gurjunene (8.6%), and γ-himachalene (25.2%). The differences in essential oil compositions may be due to genetic or environmental factors.

#### 2.1.4. *Parthenium hysterophorus*

Hydrodistillation of the fresh aerial parts of *P. hysterophorus* gave a yield of 0.05% (*w*/*w*) as a colorless/pale yellow essential oil. Gas chromatography–mass spectral analysis of the essential oil revealed a total of 75 identified (97.8% of the total) compounds (see Table 4).

The major components in the *P. hysterophorus* essential oil were germacrene D (23.2%), myrcene (14.4%), (*E*)-β-caryophyllene (12.6%), cogeijerene (4.8%), (*E*,*E*)-α-farnesene (3.3%), (*E*)-β-ocimene (3.1%), and β-pinene (3.0%). Though most of these compounds are commonly present in essential oils, cogeigerene (1,2,3,7,8,8a-hexahydro-4,8a-dimethylnaphthalene) is a relatively rare component of essential oils. The compound was originally isolated and characterized from *Geijera parviflora* [50], but it has also been found in the essential oils of *Geijera parviflora* (4.3%) [51], *Scaligeria tripartita* (1.0%) [52], and *Artemesia annua* (0.1%) [53]. The essential oil composition is qualitatively similar to an essential oil sample from Lavras, Minas Gerais, Brazil, with germacrene D (35.9%), myrcene (7.6%), (*E*)-β-caryophyllene (3.1%), (*E*)-β-ocimene (8.5%), and β-pinene (7.6%) [54]. However, neither cogeijerene nor (*E*,*E*)-α-farnesene were reported from the Brazilian sample.

#### 2.1.5. *Sphaeranthus africanus*

The essential oil from the aerial parts of *S. africanus* was obtained at 0.25% yield. The major components in *S. africanus* essential oil were 1-decen-3-ol (36.9%), α-pinene (21.0%), τ-cadinol (7.5%), 3-octyl propionate (5.6%), and (*E*)-β-caryophyllene (5.5%) (see Table 5). In contrast, the *S. africanus* (as *S. indicus*) essential oil from India was composed of thymohydroquinone dimethyl ether (18.2%), α-agarofuran (11.8%), 10-*epi*-γ-eudesmol (7.9%), and selin-11-en-4α-ol (12.7%) [55]. The compositional differences in the essential oils from Vietnam and India may be attributed to genetic differences or environmental factors.

### 2.2. Mosquito Larvicidal Activity

The essential oils of *B. lacera*, *B. sinuata*, *E. sonchifolia*, *P. hysterophorus*, and *S. africanus* were screened for mosquito larvicidal activity against *Aedes aegypti* (the yellow fever mosquito) and *Aedes albopictus* (the Asian tiger mosquito), as previously described [34,56]. The essential oils were also screened for possible insecticidal activity against the non-targeted water bug, *D. rusticus*, as previously reported [33,36]. The larvicidal and insecticidal activities for the essential oils are summarized in Table 6.

According to Dias and Moraes [39], essential oils and their components are considered to be active with larvicidal LC_50_ values less than 100 μg/mL. However, we have recently amended the activity definition: essential oils with 24-h LC_50_ < 10 μg/mL are considered “exceptionally active”, those with 24-h LC_50_ between 10 and 50 μg/mL are “very active”, those with 24-h LC_50_ between 50 and 100 μg/mL are “moderately active”, and LC_50_ >100 μg/mL are “inactive” [58]. Thus, *B. lacera* leaf essential oil was only marginally active against *Ae. aegypti* and inactive against *Ae. albopictus*.

The essential oil of *B. sinuata*, on the other hand, showed very good *Aedes* larvicidal activities with 24-h LC_50_ values of 23.4 and 29.1 μg/mL against *Ae. aegypti* and *Ae. albopictus*, respectively, as well as 48-h LC_50_ values of 17.4 and 12.4 μg/mL. Importantly, *B. sinuata* essential oil showed no mortality at the highest concentration tested (100 ug/mL) against the non-target water bug, *Diplonychus rusticus*. The larvicidal activities observed can be partly attributed to the major components. *Ayapana triplinervis* essential oil, rich in thymohydroquinone dimethyl ether (84.5%), showed larvicidal activity against *Ae. aegypti* (24-h LC_50_ = 86.2 μg/mL) [59]. (*E*)-β-Caryophyllene has shown insecticidal activity against *Ae. aegypti* larvae (LC_50_ 39–88 μg/mL), as well as *Ae. albopictus* larvae (LC_50_ 40–45 μg/mL) [33,35,36]. Likewise, α-pinene has demonstrated larvicidal activities against both *Ae. aegypti* and *Ae. albopictus* with LC_50_ values ranging 40–65 and 29–69 μg/mL, respectively [35], germacrene D showed good larvicidal activity on *Ae. aegypti* (LC_50_ = 18.8 μg/mL) [60], and α-humulene was larvicidal with 24-h LC_50_ values of 44.4 and 43.9 μg/mL against *Ae. aegypti* and *Ae. albopictus*, respectively [36].

Although *E. sonchifolia* essential oil showed moderately active mosquito larvicidal activity (24-h LC_50_ = 30.1 and 29.6 μg/mL against *Ae. aegypti* and *Ae. albopictus*, respectively), it was also insecticidal to the non-target insect, *Diplonychus rusticus* with a 24-h LC_50_ of 48.1 μg/mL. Thus, the *E. sonchifolia* essential oil is not selectively toxic and should not be considered further for this purpose.

The *Parthenum hysterophorus* essential oil showed good mosquito larvicidal activity with 24-h LC_50_ values of 47.6 and 44.4 μg/mL against *Ae. aegypti* and *Ae. albopictus*, respectively. Notably, the essential oil showed no lethality to the non-target insect, *Diplonychus rusticus*. Several of the major components of the *P. hysterophorus* essential oil have previously shown larvicidal activity against *Ae. aegypti*, including germacrene D (LC_50_ = 18.8 μg/mL) [60], myrcene (LC_50_ = 35.8 μg/mL) [61], (*E*)-β-caryophyllene (LC_50_ = 61.1 μg/mL) [36], and β-pinene (22.9 μg/mL) [33]. Larvicidal activity on *Ae. albopictus* has also been reported for myrcene [61] and (*E*)-β-caryophyllene [33] (LC_50_ = 27.0 and 56.9 μg/mL, respectively). The larvicidal activities of the major components, therefore, likely account for the observed larvicidal activities of the *P. hysterophorus* essential oil.

The essential oil of *S. africanus* showed moderate larvicidal activity with 24-h LC_50_ values of 50.7 and 36.9 μg/mL, respectively, on *Ae. aegypti* and *Ae. albopictus*. In a previous study, the *S. africanus* (as *S. indicus*) essential oil from India was screened for mosquito larvicidal activity against *Culex quinquefasciatus* and *Ae. aegypti* [62]. The larvicidal activities were very modest, however (24-h LC_50_ = 130 and 140 μg/mL, respectively). Unfortunately, the essential oil characterization in this study is not reliable.

## 3. Materials and Methods

### 3.1. Plant Material

The details of plant material collection and hydrodistillation are summarized in Table 7. During this process, botanical identification and confirmation was conducted by Dr. Huong, L.T., Faculty of Biology, College of Natural Science Education, Vinh University, Vietnam. In addition, voucher specimens with codes LTH 881, LTH 284, LTH 286, LTH 327, and LTH 332 were preserved in the plant specimen room, Vinh University, Vietnam. Aerial parts were shredded and hydrodistilled for 5 h using a Clevenger-type apparatus (Witeg Labortechnik, Wertheim, Germany). Essential oil isolation yields of three consecutive replicates were used to calculate the average yield. The essential oils were dried over anhydrous Na_2_SO_4_ and stored in sealed glass vials at 4 °C until use in analysis and bioactivity assays.

### 3.2. Gas Chromatography–Mass Spectral Analysis

Gas chromatography–mass spectral analyses (GC–MS) of *B. lacera*, *B. sinuata*, *E. sonchifolia*, *P. hysterophorus*, and *S. africanus* essential oils were carried out using the instrumentation and protocols previously published [36,56,63]. A Shimadzu GCMS-QP2010 Ultra, with a ZB-5 ms fused silica capillary column (60 m length, 0.25 mm diameter, 0.25 μm film thickness) was used, He carrier gas, 2.0 mL/min flow rate, injection and ion source temperatures of 260 °C, and a GC oven program of 50 to 260 °C at 2.0 °C/min. Injection volumes of 0.1 μL of 5% (*w*/*v*) samples of essential oil in CH_2_Cl_2_ were injected in split mode, with a 24.5:1 split ratio. Identification of the essential oil components was carried out with a comparison of MS fragmentation and retention indices (RI) with those available in the databases [64,65,66,67]. The peak areas were corrected for response using external standards of representative compounds from each compound class.

### 3.3. Mosquito Larvicidal Activity Screening

Mosquito larvicidal activity screening against *Ae. aegypti* and *Ae. albopictus* was carried out as previously described [34,56]. Quadruplicate assays using 20 fourth-instar mosquito larvae and five essential oil concentrations (100, 75, 50, 25, and 12.5 μg/mL) and a permethrin positive control. Mortality was recorded after 24 h and again after 48 h of exposure. Lethality data were subjected to log-probit analysis to obtain LC_50_ values, LC_90_ values, and 95% confidence limits using Minitab^®^ version 19.2020.1 (Minitab, LLC, State College, PA, USA).

### 3.4. Diplonychus Rusticus Insecticidal Assay

Insecticidal activity against *D. rusticus* was carried out as previously described [33]. Quadruplicate assays were conducted, using 20 adult *D. rusticus*, and five essential oil concentrations (100, 75, 50, 25, and 12.5 μg/mL), with mortality recorded after 24 h and 48 h exposure times.

## 4. Conclusions

The essential oils of *B. sinuata*, rich in thymohydroquinone dimethyl ether, (*E*)-β-caryophyllene, α-pinene, and germacrene D; *P. hysterophorus*, rich in germacrene D, myrcene, and (*E*)-β-caryophyllene; and *S. africanus*, dominated by 1-decen-3-ol and α-pinene, all showed good mosquito larvicidal activities without toxicity to a non-target aquatic species. Based on these encouraging results, *B. sinuata*, *P. hysterophorus*, and *S. africanus* essential oils should be further investigated for use as eco-friendly botanical pesticides. Field trials and formulations are needed to enhance the environmental lifetime of the essential oils and determine whether they are a viable alternative pest-control agents in aquatic systems.

## Figures and Tables

**Table 1 molecules-27-07961-t001:** Chemical compositions of *Blumea lacera* essential oils.

RI_calc_	RI_db_	Compound	%		
			Floral	Leaf	Stem
931	933	α-Pinene	0.5	0.1	0.1
949	950	Camphene	tr	tr	---
971	972	Sabinene	0.1	tr	---
990	986	Safranal	0.1	0.4	tr
1024	1025	*p*-Cymene	0.1	tr	tr
1028	1030	Limonene	tr	tr	---
1057	1054	γ-Terpinene	tr	tr	---
1063	1086	2,6,6-Trimethyl-1,4-cyclohexadiene-1-carboxaldehyde	0.1	tr	tr
1099	1101	Linalool	0.2	0.1	tr
1101	1106	Filifolone	1.2	0.9	0.1
1105	1107	Nonanal	0.1	0.1	---
1108	1106	*iso*-Chrysanthenone	0.1	0.1	---
1112	1110	(*E*)-4,8-Dimethylnona-1,3,7-triene	0.1	tr	tr
1122	1124	Chrysanthenone	1.3	1.0	0.1
1129	1129	1,3,8-*p*-Menthatriene	---	---	0.1
1137	1136	*trans*-Chrysanthenol	0.3	0.2	0.1
1159	1152	Albene	tr	tr	0.2
1223	1215	Isothymyl methyl ether	---	---	tr
1229	1229	Thymyl methyl ether	0.1	0.1	1.0
1238	1239	Carvacryl methyl ether	tr	tr	0.1
1256	1261	*cis*-Chrysanthenyl acetate	0.1	---	---
1284	1285	Bornyl acetate	---	---	0.1
1290	1289	Thymol	---	---	tr
1345	1345	Silphinene	---	---	tr
1374	1375	α-Copaene	0.1	0.1	tr
1380	1381	*cis*-β-Elemene	tr	0.1	0.1
1382	1382	β-Bourbonene	tr	tr	tr
1386	1387	β-Cubebene	0.1	0.1	tr
1388	1390	*trans*-β-Elemene	1.6	2.1	1.1
1394	1392	2-Ethylidene-6-methyl-3,5-heptadienal	0.9	0.6	---
1413	1411	Thymohydroquinone dimethyl ether	5.0	4.1	28.4
1419	1417	(*E*)-β-Caryophyllene	23.8	27.2	11.7
1428	1430	β-Copaene	0.2	0.1	0.1
1431	1432	*trans*-α-Bergamotene	0.3	0.2	0.2
1434	1443	Dimethoxy-*p*-cymenene	0.1	tr	0.5
1445	1446	*epi*-β-Santene	---	---	0.1
1446	1453	Geranyl acetone	0.1	tr	tr
1451	1452	(*E*)-β-Farnesene	0.9	0.8	0.5
1455	1454	α-Humulene	3.7	3.5	1.5
1459	1457	*allo*-Aromadendrene	0.1	0.1	tr
1461	1461	*cis*-Cadina-1(6),4-diene	0.1	0.1	---
1471	1475	*trans*-Cadina-1(6),4-diene	tr	---	---
1473	---	Unidentified (43, 148, 218)	0.4	0.2	1.0
1475	1480	Thymyl isobutyrate	0.9	0.5	2.7
1477	1481	γ-Curcumene	5.9	7.7	4.7
1480	1479	*ar*-Curcumene	8.0	3.7	1.9
1482	1483	Germacrene D	18.5	21.0	11.2
1482	1490	Neryl isobutyrate	tr	---	5.2
1488	1489	β-Selinene	0.4	0.3	0.2
1491	1492	*trans*-Muurola-4(14),5-diene	tr	0.1	0.1
1493	1493	α-Zingiberene	5.7	7.1	4.6
1497	1497	α-Muurolene	0.5	0.4	0.2
1502	1504	(*E*,*E*)-α-Farnesene	0.6	0.2	0.1
1506	1508	β-Bisabolene	tr	0.1	0.1
1512	1514	γ-Cadinene	0.5	0.3	0.3
1516	1518	δ-Cadinene	1.7	1.0	0.6
1522	1521	β-Sesquiphellandrene	3.6	3.8	2.4
1559	1562	(*E*)-Nerolidol	0.4	0.2	0.1
1565	1571	Thymyl 2-methylbutanoate	0.9	0.4	1.1
1568	1571	Neryl 2-methylbutanoate	0.9	0.8	3.3
1576	1580	Neryl isovalerate	0.8	0.5	1.3
1582	1587	Caryophyllene oxide	1.4	1.4	0.5
1598	1598	Humulene epoxide I	0.2	0.2	0.1
1610	1611	Humulene epoxide II	0.1	0.1	tr
1613	1611	Zingiberenol	0.3	0.2	0.3
1630	1632	7-*epi-cis*-Sesquisabinene hydrate	0.2	0.1	0.3
1634	1635	Caryophylla-4(12),8(13)-dien-5α-ol	0.1	0.1	0.1
1637	1636	Caryophylla-4(12),8(13)-dien-5β-ol	0.1	0.3	0.1
1639	1638	(2*S*,5*E*)-Caryophyll-5-en-12-al	1.0	1.1	0.4
1642	1643	τ-Cadinol	0.8	0.8	3.0
1644	1645	τ-Muurolol	0.4	0.3	0.4
1646	1645	δ-Cadinol	0.1	0.1	0.1
1651	1650	β-Eudesmol	---	---	0.1
1655	1655	α-Cadinol	1.2	1.0	1.3
1659	1658	Selin-11-en-4α-ol	tr	0.1	0.1
1665	1665	Intermedeol	tr	---	0.1
1667	1673	6-Methoxythymyl isobutyrate	0.2	0.1	0.9
1670	1671	14-Hydroxy-9-*epi*-(*E*)-caryophyllene	---	0.1	---
1685	1687	4-Himachalen-1β-ol (2-Himachalen-6β-ol)	0.7	0.5	0.8
1687	1685	α-Bisabolol	0.1	0.1	---
1694	1713	(2Z,6Z)-Farnesal	0.1	---	---
1713	1715	Pentadecanal	---	0.2	0.2
1745	1751	Xanthorrhizol	---	---	0.1
1780	1780	(*Z*)-Nerolidyl isobutyrate	0.2	---	---
2013	---	Unidentified (43, 71, 145, 162)	0.3	0.2	1.8
2098	---	Unidentified (43, 57, 71, 85, 145, 162)	0.5	0.3	1.0
2103	2106	(*E*)-Phytol	tr	0.2	0.1
2500	2500	Pentacosane	tr	tr	0.1
		Monoterpene hydrocarbons	0.6	0.1	0.1
		Oxygenated monoterpenoids	13.0	9.3	44.9
		Sesquiterpene hydrocarbons	76.1	80.1	41.5
		Oxygenated sesquiterpenoids	7.6	6.8	7.7
		Others	0.4	0.9	0.6
		Total identified	97.7	97.3	94.7

RI_calc._ = Retention indices determined with reference to a homologous series of *n*-alkanes on a ZB-5 ms column. RI_db_ = Retention indices from the databases. tr = trace (<0.05%). % = percent of total essential oil composition.

**Table 2 molecules-27-07961-t002:** Essential oil composition of *Blumea sinuata* from Vietnam.

RI_calc_	RI_db_	Compound	%
925	925	α-Thujene	tr
933	933	α-Pinene	8.8
949	950	Camphene	tr
952	953	Thuja-2,4(10)-diene	tr
970	969	Dimethyltrisulfide	tr
972	972	Sabinene	tr
977	978	β-Pinene	tr
985	986	6-Methylhept-5-en-2-one	tr
988	989	Myrcene	0.1
989	989	2-Pentylfuran	tr
1007	1007	α-Phellandrene	tr
1025	1025	*p*-Cymene	0.1
1029	1030	Limonene	0.1
1031	1031	β-Phellandrene	tr
1045	1045	(*E*)-β-Ocimene	tr
1100	1101	Linalool	0.1
1106	1107	Nonanal	0.1
1107	1107	1-Octen-3-yl acetate	tr
1110	1108	*p*-Mentha-2,8-dien-1-ol	tr
1113	1113	(*E*)-1,5-Dimethylnona-1,3,7-triene	tr
1146	1145	*trans*-Verbenol	tr
1159	1161	Albene	0.3
1196	1195	α-Terpineol	tr
1207	1206	Decanal	tr
1230	1229	Thymyl methyl ether	0.4
1239	1239	Carvacryl methyl ether	tr
1266	1272	Nonanoic acid	0.1
1284	1285	Bornyl acetate	0.3
1323	1326	Myrtenyl acetate	tr
1346	1348	α-Cubebene	0.1
1350	1348	α-Longipinene	tr
1359	1361	Neryl acetate	0.3
1371	1371	Decanoic acid	1.5
1375	1375	α-Copaene	1.2
1383	1382	β-Bourbonene	0.1
1387	1387	β-Cubebene	0.4
1389	1390	*trans*-β-Elemene	0.3
1415	1411	Thymohydroquinone dimethyl ether	29.4
1420	1417	(*E*)-β-Caryophyllene	19.7
1430	1430	β-Copaene	0.2
1433	1432	*trans*-α-Bergamotene	0.1
1441	1439	(*Z*)-β-Farnesene	0.1
1447	1446	*epi*-β-Santalene	0.1
1453	1452	(*E*)-β-Farnesene	3.5
1456	1454	α-Humulene	4.3
1460	1457	*allo*-Aromadendrene	0.5
1475	1478	γ-Muurolene	0.1
1479	1481	(*E*)-β-Ionone	0.1
1482	1483	Germacrene D	7.8
1484	1483	*trans*-β-Bergamotene	0.5
1489	1489	β-Selinene	0.1
1492	1492	*trans*-Muurola-4(14),5-diene	0.1
1496	1497	α-Selinene	0.7
1498	1497	α-Muurolene	0.2
1504	1504	(*E*,*E*)-α-Farnesene	1.1
1508	1508	β-Bisabolene	0.1
1513	1514	γ-Cadinene	0.1
1518	1515	Dihydrolachnophyllum ester B	1.0
1518	1518	δ-Cadinene	0.8
1522	1519	*trans*-Calamenene	0.1
1524	1523	7-*epi-cis*-Sesquisabinene hydrate	0.2
1561	1562	(*E*)-Nerolidol	0.4
1564	1561	7-Hydroxyfarnesene	0.2
1571	1568	Palustrol	0.2
1579	1580	Neryl isovalerate	0.6
1583	1587	Caryophyllene oxide	3.6
1593	1593	Salvial-4(14)-en-1-one	0.1
1605	1605	Ledol	0.2
1611	1611	Humulene epoxide II	0.4
1613	1610	(*Z*)-Sesquilavandulol	0.2
1617	1611	β-Atlantol	0.2
1629	1628	1-*epi*-Cubenol	0.1
1635	1635	Caryophylla-4(12),8(13)-dien-5α-ol	0.1
1638	1636	Caryophylla-4(12),8(13)-dien-5β-ol	0.5
1640	1639	*allo*-Aromadendrene epoxide	0.1
1643	1643	τ-Cadinol	0.3
1645	1645	τ-Muurolol	0.2
1647	1653	Pogostol	0.2
1656	1655	α-Cadinol	0.8
1671	1671	14-Hydroxy-9-*epi*-(*E*)-caryophyllene	0.3
1680	1683	15-Hydroxy-α-muurolene	0.3
1686	1683	Germacra-4(15),5,10(14)-trien-1α-ol	0.5
1716	1715	Pentadecanal	0.4
1841	1841	Phytone	0.1
1862	1856	(*Z*)-Lanceol acetate	2.6
		Monoterpene hydrocarbons	9.1
		Oxygenated monoterpenoids	30.5
		Sesquiterpene hydrocarbons	42.4
		Oxygenated sesquiterpenoids	12.2
		Others	3.5
		Total identified	97.8

RI_calc._ = Retention indices determined with reference to a homologous series of *n*-alkanes on a ZB-5 ms column. RI_db_ = Retention indices from the databases. tr = trace (<0.05%). % = percent of total essential oil composition.

**Table 3 molecules-27-07961-t003:** Essential oil composition of *Emilia sonchifolia* from Vietnam.

RI_calc_	RI_db_	Compound	%
882	880	2-Butylfuran	0.3
933	932	α-Pinene	2.4
949	950	Camphene	0.2
977	978	β-Pinene	1.2
989	989	Myrcene	0.8
991	987	1-Decene	0.4
1024	1024	*p*-Cymene	1.7
1029	1030	Limonene	1.5
1046	1045	(*E*)-β-Ocimene	0.8
1092	1091	1-Undecene	41.9
1335	1335	δ-Elemene	0.6
1369	1367	Cyclosativene	0.3
1375	1375	α-Copaene	0.3
1387	1387	β-Cubebene	0.4
1389	1390	*trans*-β-Elemene	1.4
1418	1417	(*E*)-β-Caryophyllene	2.2
1428	1427	γ-Elemene	0.6
1452	1452	(*E*)-β-Farnesene	0.2
1454	1454	α-Humulene	2.8
1459	1461	Precocene I (=6-Demethoxyageratochromene)	0.8
1474	1475	γ-Muurolene	0.6
1480	1480	Germacrene D	11.0
1492	1492	1-Pentadecene	0.2
1497	1497	α-Muurolene	0.5
1503	1503	(*E*,*E*)-α-Farnesene	0.3
1506	1508	β-Bisabolene	1.4
1511	1512	γ-Cadinene	0.4
1517	1518	δ-Cadinene	0.8
1527	1528	Kessane	0.5
1557	1557	Germacrene B	0.6
1559	1561	(*E*)-Nerolidol	1.1
1566	1566	1,5-Epoxysalvial-4(14)-ene	0.9
1575	1576	Spathulenol	1.0
1580	1577	Caryophyllene oxide	1.3
1607	1607	Humulene epoxide I	1.2
1626	1629	*iso*-Spathulenol	0.7
1637	1644	*allo*-Aromadendrene epoxide	0.8
1640	1640	τ-Cadinol	0.3
1642	1644	τ-Muurolol	0.5
1653	1655	α-Cadinol	3.8
1659	---	Unidentified (43, 79, 91, 105, 133(100%), 163, 206)	1.1
1666	---	Unidentified (41, 55, 81(100%), 93, 164, 206)	1.2
1827	---	Unidentified (41, 55, 81, 123(100%), 151, 191)	2.8
1839	1841	Phytone	0.8
2113	2109	Phytol	3.8
		Monoterpene hydrocarbons	8.7
		Oxygenated monoterpenoids	0.0
		Sesquiterpene hydrocarbons	24.7
		Oxygenated sesquiterpenoids	11.6
		Diterpenoids	3.8
		Others	44.4
		Total identified	93.2

RI_calc._ = Retention indices determined with reference to a homologous series of *n*-alkanes on a ZB-5 ms column. RI_db_ = Retention indices from the databases. tr = trace (<0.05%). % = percent of total essential oil composition.

**Table 4 molecules-27-07961-t004:** Essential oil composition of *Parthenium hysterophorus* from Vietnam.

RI_calc_	RI_db_	Compound	%
922	923	Tricyclene	0.1
925	925	α-Thujene	tr
932	932	α-Pinene	1.0
949	950	Camphene	2.2
972	972	Sabinene	0.6
978	978	β-Pinene	3.0
979	978	1-Octen-3-ol	0.3
986	986	Octan-3-one	tr
990	989	Myrcene	14.4
1025	1025	*p*-Cymene	0.1
1030	1030	Limonene	1.0
1031	1031	β-Phellandrene	0.5
1036	1035	(*Z*)-β-Ocimene	tr
1046	1046	(*E*)-β-Ocimene	3.1
1052	1051	2,3,6-Trimethylhepta-1,5-diene	0.1
1058	1058	γ-Terpinene	tr
1081	1079	1-Nonen-3-ol	0.2
1086	1086	Terpinolene	tr
1099	1098	Perillene	0.1
1101	1101	Linalool	0.1
1114	1114	4,8 Dimethylnona-1,3,7-triene	0.4
1140	1139	(*E*)-Myroxide	tr
1182	1180	Terpinen-4-ol	0.1
1189	1187	Cryptone	tr
1286	1286	Cogeijerene	4.8
1332	1331	Bicycloelemene	0.1
1335	1335	δ-Elemene	0.3
1347	1348	α-Cubebene	0.1
1370	1367	Cyclosativene	0.2
1376	1375	α-Copaene	0.3
1379	1380	Daucene	0.2
1382	1383	*cis*-β-Elemene	0.4
1384	1385	β-Bourbonene	0.5
1388	1387	β-Cubebene	0.7
1390	1390	*trans*-β-Elemene	0.9
1392	1392	Sativene	0.1
1416	1414	α-Cedrene	0.1
1421	1418	(*E*)-β-Caryophyllene	12.6
1430	1432	γ-Elemene	0.7
1433	1432	*trans*-α-Bergamotene	0.1
1441	1439	(*Z*)-β-Farnesene	0.1
1442	1442	Guaia-6,9-diene	0.1
1445	1447	*iso*-Germacrene D	0.1
1454	1452	(*E*)-β-Farnesene	0.2
1456	1454	α-Humulene	1.5
1476	1478	γ-Muurolene	2.5
1484	1483	Germacrene D	23.2
1490	1489	β-Selinene	0.2
1493	1492	*trans*-Muurola-4(15),5-diene	0.1
1496	1497	Bicyclogermacrene	0.8
1499	1500	α-Muurolene	0.5
1505	1504	(*E*,*E*)-α-Farnesene	3.3
1508	1508	β-Bisabolene	0.1
1514	1514	γ-Cadinene	0.1
1516	1515	Cubebol	0.2
1519	1520	δ-Cadinene	0.6
1525	1524	β-Sesquiphellandrene	0.2
1533	1532	Selina-4(15),7(11)-diene	0.4
1560	1560	Germacrene B	0.4
1562	1560	(*E*)-Nerolidol	0.6
1566	1571	*iso*-Shyobunol	2.8
1578	1576	Spathulenol	0.5
1584	1587	Caryophyllene oxide	2.4
1604	1609	Carotol	1.8
1611	1611	Humulene epoxide II	0.2
1628	1624	Muurola-4,10(14)-dien-1α-ol	0.6
1630	1629	*iso*-Spathulenol	0.3
1634	1632	Muurola-4,10(14)-dien-1β-ol	1.4
1641	1644	*allo*-Aromadendrene epoxide	0.7
1644	1643	τ-Cadinol	0.1
1646	1645	τ-Murrolol	0.1
1648	1651	α-Muurolol (=δ-Cadinol)	0.6
1657	1655	α-Cadinol	0.6
1865	1860	Platambin	0.3
2109	2109	Phytol	0.5
		Monoterpene hydrocarbons	26.1
		Oxygenated monoterpenoids	0.2
		Sesquiterpene hydrocarbons	51.9
		Oxygenated sesquiterpenoids	13.2
		Diterpenoids	0.5
		Others	5.7
		Total identified	97.8

RI_calc._ = Retention indices determined with reference to a homologous series of *n*-alkanes on a ZB-5 ms column. RI_db_ = Retention indices from the databases. tr = trace (<0.05%). % = percent of total essential oil composition.

**Table 5 molecules-27-07961-t005:** Essential oil composition of *Sphaeranthus africanus* from Vietnam.

RI_calc_	RI_db_	Compound	%
926	925	α-Thujene	tr
934	933	α-Pinene	21.0
950	950	Camphene	0.1
953	953	Thuja-2,4(10)-diene	tr
973	972	Sabinene	0.1
978	978	β-Pinene	0.2
979	982	1-Octen-3-ol	0.2
989	989	Myrcene	0.1
1025	1025	*p*-Cymene	0.2
1029	1030	Limonene	0.1
1046	1045	(*E*)-β-Ocimene	0.2
1081	1079	1-Nonen-3-ol	0.1
1099	1099	(2*Z*)-Hexenyl propanoate	0.9
1106	1107	Nonanal	0.1
1108	1107	1-Octen-3-yl acetate	0.7
1111	1109	Vinyl 2-ethylhexanoate	0.3
1120	1118	3-Octyl acetate	0.4
1194	1184	1-Decen-3-ol	36.9
1205	1218	3-Octyl propionate	5.6
1216	1218	3-Nonyl acetate	0.1
1229	1229	Thymyl methyl ether	0.2
1242	1242	Cuminaldehyde	0.1
1250	1249	6-Methyldodecane	0.2
1290	1294	2,2,4,4,6,8,8-Heptamethylnonane	2.6
1295	1294	*trans*-Pinocarvyl acetate	0.1
1322	1322	Myrtenyl acetate	0.1
1345	1349	7-*epi*-Silphiperfol-5-ene	0.3
1380	1382	Modheph-2-ene	2.4
1387	1385	α-Isocomene	0.4
1409	1413	β-Isocomene	0.4
1411	1411	Thymohydroquinone dimethyl ether	0.4
1418	1417	(*E*)-β-Caryophyllene	5.5
1452	1452	(*E*)-β-Farnesene	0.1
1454	1454	α-Humulene	0.4
1458	1458	*allo*-Aromadendrene	0.5
1460	1461	Precocene 1 (=6-Demethoxyageratochromene)	0.5
1479	1480	Germacrene D	0.1
1496	1497	α-Muurolene	0.1
1511	1512	γ-Cadinene	0.8
1515	1518	Isoshyobunone	0.5
1516	1518	δ-Cadinene	0.3
1579	1577	Caryophyllene oxide	1.1
1595	1597	Dimethyl-α-ionone	0.2
1600	1600	β-Oplopenone	0.1
1601	1604	Geranyl isovalerate	0.1
1623	1624	Muurola-4,10(14)-dien-1β-ol	0.1
1631	1631	Caryophylla-4(12),8(13)-dien-5α-ol	0.1
1634	1636	Caryophylla-4(12),8(13)-dien-5β-ol	0.2
1640	1641	τ-Cadinol	7.5
1651	1652	β-Himachalol	1.5
1662	1660	Selin-11-en-4β-ol	0.1
1671	1672	Jatamansone	2.0
1834	1836	Neophytadiene	0.4
1839	1841	Phytone	0.3
2103	2102	Phytol	2.0
		Monoterpene hydrocarbons	22.0
		Oxygenated monoterpenoids	1.1
		Sesquiterpene hydrocarbons	10.9
		Oxygenated sesquiterpenoids	13.4
		Diterpenoids	2.3
		Others	48.9
		Total identified	98.5

RI_calc._ = Retention indices determined with reference to a homologous series of *n*-alkanes on a ZB-5 ms column. RI_db_ = Retention indices from the databases. tr = trace (<0.05%). % = percent of total essential oil composition.

**Table 6 molecules-27-07961-t006:** *Aedes* larvicidal and *Diplonychus rusticus* insecticidal activities of Vietnamese Asteraceae essential oils.

	** *Aedes Aegypti* **			
**Essential Oil**	**24 h**		**48 h**	
	LC_50_	LC_90_	LC_50_	LC_90_
*Blumea lacera* leaf	64.7 (59.8–70.1)	96.4 (89.4–105.3)	55.1 (50.5–60.2)	83.4 (76.6–92.1)
*Blumea sinuata* aerial parts	23.4 (21.2–25.8)	36.2 (32.6–41.9)	17.4 (15.6–19.1)	27.3 (24.8–31.3)
*Emilia sonchifolia* aerial parts	30.1 (27.9–32.9)	40.8 (37.3–46.0)	26.2 (24.2–28.8)	36.6 (33.1–42.1)
*Parthenium hysterophorus* aerial parts	47.6 (44.7–50.5)	63.4 (59.7–68.5)	36.3 (33.2–39.6)	57.7 (53.1–63.9)
*Sphaeranthus africanus* aerial parts	50.7 (46.6–55.9)	74.4 (67.4–84.6)	44.2 (40.8–48.4)	65.3 (59.4–73.6)
	*Aedes albopictus*			
	24 h		48 h	
	LC_50_	LC_90_	LC_50_	LC_90_
*Blumea lacera* leaf	116.7 (110.3–123.7)	155.8 (146.4–168.5)	99.4 (92.5–107.0)	147.4 (136.8–161.3)
*Blumea sinuata* aerial parts	29.1 (24.7–33.4)	104.7 (85.0–239.3)	12.4 (9.6–14.9)	36.5 (31.0–45.5)
*Emilia sonchifolia* aerial parts	29.6 (27.4–32.0)	46.3 (42.8–50.9)	23.4 (21.3–25.7)	40.7 (37.2–45.5)
*Parthenium hysterophorus* aerial parts	44.4 (41.2–47.8)	66.4 (61.7–72.4)	33.8 (29.9–37.6)	63.6 (57.9–71.2)
*Sphaeranthus africanus* aerial parts	36.9 (34.3–39.6)	56.4 (52.5–61.3)	28.8 (26.6–31.2)	44.4 (40.9–49.0)
	*Diplonychus rusticus*			
	24 h		48 h	
	LC_50_	LC_90_	LC_50_	LC_90_
*Blumea lacera* leaf	>50	>50	>50	>50
*Blumea sinuata* aerial parts	>100	>100	>100	>100
*Emilia sonchifolia* aerial parts	48.1 (±8.9) ^a^	---	34.4 (±8.9) ^a^	---
*Parthenum hysterophorus* aerial parts	>100	>100	>100	>100
*Sphaeranthus africanus* aerial parts	>50	>50	>50	>50

^a^ Due to insufficient data for probit analysis, the LC_50_ was determined using the Reed–Muench method [57].

**Table 7 molecules-27-07961-t007:** Details for collecting essential oils of *Blumea lacera*, *Blumea sinuata*, *Emilia sonchifolia*, *Parthenium hysterophorus*, and *Sphaeranthus africanus*.

Plant Species	Collection Location (GPS)	Part	Mass Plant Material (kg)	Extraction Yield (%*w*/*w*)	Collection Time
*B. lacera*	Nghia Dan District, Nghe An Province (19°23′05″ N, 105°25′51″ E).	Aerial parts	3.0	1.2	August 2021
		Leaves	0.3	1.56	August 2021
		Flowers	0.3	1.10	August 2021
		Stems	0.3	0.35	August 2021
*B. sinuata*	Nghia Dan District, Nghe An Province (19°20′06″ N, 105°25′59″ E).	Aerial parts	4.0	0.16	August 2021
*E. sonchifolia*	Diên Lãm Commune, Pù Huống Natural Reserve, Nghệ An Province (19°26′44″ N, 104°58′40″ E).	Aerial parts	3.0	0.51	August 2021
*P. hysterophorus*	Bình Chuẩn Commune, Pù Huống Natural Reserve, Nghệ An Province (19°16′53″ N, 104°55′16″ E).	Aerial parts	5.0	0.05	August 2021
*S. africanus*	Diên Lãm Commune, Pù Huống Natural Reserve, Nghệ An Province (19°26′44″ N, 104°58′40″ E).	Aerial parts	4.0	0.25	August 2021

## Data Availability

All data are available upon reasonable request from the corresponding authors (N.H.H. and W.N.S.).

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
