# Peer review of "Essential Oils from Vietnamese Asteraceae for Environmentally Friendly Control of Aedes Mosquitoes"

_molecules, 2022, doi:10.3390/molecules27227961_

Round 1
Reviewer 1 Report
The questions and comments are inserted in pdf file by editing tools (highliting and comments).
The importance of the work should be explained in more details.
The English should be corrected and the text written more literary.
The discussion should be written more clearly, not just by listing many facts.
Lines 47-49: Instead of paragraph „The Asteraceae is the largest family of flora in the world. There are about 1550 genera and about 23,000 species [1]. In Vietnam there are about 126 genera and 379 species [2]. Many species are used as medicines, for essential oils, or as ornamentals [2]“. write this one:
“The Asteraceae is the largest family of flora in the world comprising about 1550 genera and about 23,000 species [1]. In Vietnam there are about 126 genera and 379 species in this family [2]. Many species are used as medicines, for isolation of essential oils, or as ornamentals [2]”
Line 88: delete “for”
Line 106: instead of “give” write “obtain”
Reviewer 2 Report
The authors presented a v good well written
The manuscript is not in nee manuscriptd to be revised and i accept its publication
Just a simple comment which apha, beta, gama,...symbols should be in italic
Reviewer 3 Report
I commend the authors Tran Minh Hoi , Prabodh Satyal , Le Thi Huong , Dang Viet Hau, Tran Duc Binh, Dang Thi Hong Duyen, Do Ngoc Dai, Ngo Gia Huy, Hoang Van Chinh , Vo Van Hoa , Nguyen Huy Hung, William N. Setzer of the manuscript titled “Essential Oils from Vietnamese Asteraceae for Environmentally-Friendly Control of Aedes Mosquitoes” for their work on the isolation Blumea lacera, Blumea sinuata, Emilia sonchifolia, Parthenium hysterophorus, and Sphaeranthus africanus essential oils for the control of Aedes aegypti and Aedes albopictus.
Before this paper is published, there are several things need to be addressed or corrected:
1- In the introduction,
- The introduction is very short and does not have enough literature data.
- Additional literature should be added about the previous work on the isolation of the essential oils and their biological activities and focus on showing the novelty wherever possible in this work.
-
2- In the results and discussion part, additional references need to be added to discuss previous publications on genetically close genera.
- The percentage of the essential oil (yield) of Blumea lacera should be mentioned.
- In Emilia sonchifolia essential oils, explain why the results are different from previous reports on the same species.
- The pervious comment applies to other studies species.
- A detailed discussion should be added about the possible active compounds responsible for the mosquito larvicidal activity also the related literature and other sources of these compounds should be discussed
3- In the materials and methods:
Further details of the analyses should be added to the material and methods parts.
Round 2
Reviewer 3 Report
accepted for me